# CHALLENGES IN DISENTANGLING INDEPENDENT FACTORS OF VARIATION

## ABSTRACT

We study the problem of building models that disentangle independent factors of variation. Such models encode features that can efficiently be used for classification and to transfer attributes between different images in image synthesis. As data we use a weakly labeled training set, where labels indicate what single factor has changed between two data samples, although the relative value of the change is unknown. This labeling is of particular interest as it may be readily available without annotation costs. We introduce an autoencoder model and train it through constraints on image pairs and triplets. We show the role of feature dimensionality and adversarial training theoretically and experimentally. We formally prove the existence of the reference ambiguity, which is inherently present in the disentangling task when weakly labeled data is used. The numerical value of a factor has different meaning in different reference frames. When the reference depends on other factors, transferring that factor becomes ambiguous. We demonstrate experimentally that the proposed model can successfully transfer attributes on several datasets, but show also cases when the reference ambiguity occurs.

## 1 INTRODUCTION

One way to simplify the problem of classifying or regressing attributes of interest from data is to build an intermediate representation, a feature, where the information about the attributes is better separated than in the input data. Better separation means that some entries of the feature vary only with respect to one and only one attribute. In this way, classifiers and regressors would not need to build invariance to many nuisance attributes. Instead, they could devote more capacity to discriminating the attributes of interest, and possibly achieve better performance. We call this task *disentangling factors of variation*, and we identify attributes with the factors. In addition to facilitating classification and regression, this task is beneficial to image synthesis. One could build a model to render images, where each input varies only one attribute of the output, and to transfer attributes between images.

When labeling is possible and available, supervised learning can be used to solve this task. In general, however, some attributes may not be easily quantifiable (*e.g.*, style). Therefore, we consider using *weak labeling*, where we only know what attribute has changed between two images, although we do not know by how much. This type of labeling may be readily available in many cases without manual annotation. For example, image pairs from a stereo system are automatically labeled with a viewpoint change, albeit unknown. A practical model that can learn from these labels is an encoder-decoder pair subject to a reconstruction constraint. In this model the weak labels can be used to define similarities between subsets of the feature obtained from two input images.

We introduce a *novel adversarial training* of autoencoders to solve the disentangling task when only weak labels are available. Compared to previous methods, our discriminator is not conditioned on class labels, but takes image pairs as inputs. This way the number of parameters can be kept constant.

We describe the *shortcut problem*, where all the the information is encoded only in one part of the feature, while other part is completely ignored, as fig. 1 illustrates. We prove our method solves this problem and demonstrate it experimentally.

We formally prove existence of the *reference ambiguity*, that is inherently present in the disentangling task when weak labels are used. Thus no algorithm can provably learn disentangling. As fig. 1 shows, the reference ambiguity means that a factor (for example viewpoint) can have different meaning when

using a different reference frame that depends on another factor (for example car type). We show experimentally that this ambiguity rarely arise, we can observe it only when the data is complex.

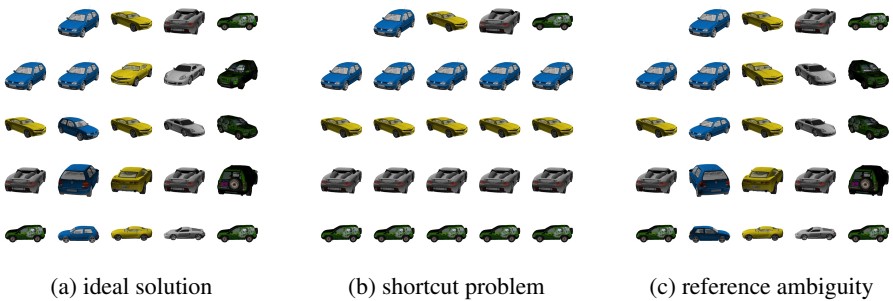

|   (a) ideal solution   |   (b) shortcut problem   |   (c) reference ambiguity   |

Figure 1: Challenges of disentangling. We disentangle the feature into two parts, one representing the viewpoint, the other the car type. We use the features for attribute transfer. For all subfigures the viewpoint feature is taken from the leftmost column and the car type feature is taken from the topmost row. (a) ideal solution: the viewpoint and the car type are transferred correctly. (b) shortcut problem: the car type is not transferred. (c) reference ambiguity: compared to the others the viewpoint orientation is flipped for the blue car.

## 2 RELATED WORK

**Autoencoders.** Autoencoders in Bourlard & Kamp (1988), Hinton & Salakhutdinov (2006), Bengio et al. (2013) learn to reconstruct the input data as $\mathbf{x} = \text{Dec}(\text{Enc}(\mathbf{x}))$, where $\text{Enc}(\mathbf{x})$ is the internal image representation (the encoder) and Dec (the decoder) reconstructs the input of the encoder. Variational autoencoders in Kingma & Welling (2014) use a generative model; $p(\mathbf{x}, \mathbf{z}) = p(\mathbf{x}|\mathbf{z})p(\mathbf{z})$, where $\mathbf{x}$ is the observed data (images), and $\mathbf{z}$ are latent variables. The encoder estimates the parameters of the posterior, $\text{Enc}(\mathbf{x}) = p(\mathbf{z}|\mathbf{x})$, and the decoder estimates the conditional likelihood, $\text{Dec}(\mathbf{z}) = p(\mathbf{x}|\mathbf{z})$. In Hinton et al. (2011) autoencoders are trained with transformed image input pairs. The relative transformation parameters are also fed to the network. Because the internal representation explicitly represents the objects presence and location, the network can learn their absolute position. One important aspect of the autoencoders is that they encourage latent representations to keep as much information about the input as possible.

**GAN.** Generative Adversarial Nets Goodfellow et al. (2014) learn to sample realistic images with two competing neural networks. The generator Dec creates images $\mathbf{x} = \text{Dec}(\mathbf{z})$ from a random noise sample $\mathbf{z}$ and tries to fool a discriminator Dsc, which has to decide whether the image is sampled from the generator $p_g$ or from real images $p_{real}$. After a successful training the discriminator cannot distinguish the real from the generated samples. Adversarial training is often used to enforce constraints on random variables. BIGAN, Donahue et al. (2016) learns a feature representation with adversarial nets by training an encoder Enc, such that $\text{Enc}(\mathbf{x})$ is Gaussian, when $\mathbf{x} \sim p_{real}$. CoGAN, Liu & Tuzel (2016) learns the joint distribution of multi-domain images by having generators and discriminators in each domain, and sharing their weights. They can transform images between domains without being given correspondences. InfoGan, Chen et al. (2016) learns a subset of factors of variation by reproducing parts of the input vector with the discriminator.

**Disentangling and independence.** Many recent methods use neural networks for disentangling features, with various degrees of supervision. In Xi Peng (2017) multi-task learning is used with full supervision for pose invariant face recognition. Using both identity and pose labels Tran et al. (2017) can learn pose invariant features and synthesize frontalized faces from any pose. In Yang et al. (2015) autoencoders are used to generate novel viewpoints of objects. They disentangle the object category factor from the viewpoint factor by using as explicit supervision signals: the relative viewpoint transformations between image pairs. In Cheung et al. (2014) the output of the encoder is split in two parts: one represents the class label and the other represents the nuisance factors. Their objective function has a penalty term for misclassification and a cross-covariance cost to disentangle class from nuisance factors. Hierarchical Boltzmann Machines are used in Reed et al. (2014) for disentangling. A subset of hidden units are trained to be sensitive to a specific factor of variation, while being

invariant to others. Variational Fair Autoencoders Louizos et al. (2016) learn a representation that is invariant to specific nuisance factors, while retaining as much information as possible. Autoencoders can also be used for visual analogy Reed et al. (2015). GAN is used for disentangling intrinsic image factors (albedo and normal map) in Shu et al. (2017) without using ground truth labeling. They achieve this by explicitly modeling the physics of the image formation in their network.

The work most related to ours is Mathieu et al. (2016), where an autoencoder restores an image from another by swapping parts of the internal image representation. Their main improvement over Reed et al. (2015) is the use of adversarial training, which allows for learning with image pairs instead of image triplets. Therefore, expensive labels like viewpoint alignment between different car types are no longer needed. One of the differences between this method and ours is that it trains a discriminator for each of the given labels. A benefit of this approach is the higher selectivity of the discriminator, but a drawback is that the number of model parameters grows linearly with the number of labels. In contrast, we work with image pairs and use a single discriminator so that our method is uninfluenced by the number of labels. Moreover, we show formally and experimentally the difficulties of disentangling factors of variation.

## 3 DISENTANGLING FACTORS OF VARIATION

We are interested in the design and training of two models. One should map a data sample (*e.g.*, an image) to a feature that is explicitly partitioned into subvectors, each associated to a specific factor of variation. The other model should map this feature back to an image. We call the first model the *encoder* and the second model the *decoder*. For example, given the image of a car we would like the encoder to yield a feature with two subvectors: one related to the car viewpoint, and the other related to the car type. The subvectors of the feature obtained from the encoder should be useful for classification or regression of the corresponding factor that they depend on (the car viewpoint and type in the example). This separation would also be very useful to the decoder. It would enable advanced editing of images, for example, the transfer of the viewpoint or car types from an image to another, by swapping the corresponding subvectors. Next, we introduce our model of the data and formal definitions of our encoder and decoder.

**Data model.** We assume that our observed data $\mathbf{x}$ is generated through some unknown deterministic invertible and smooth process $f$ that depends on the factors $\mathbf{v}$ and $\mathbf{c}$, so that $\mathbf{x} = f(\mathbf{v}, \mathbf{c})$. In our earlier example, $\mathbf{x}$ is an image, $\mathbf{v}$ is a viewpoint, $\mathbf{c}$ is a car type, and $f$ is the rendering engine. It is reasonable to assume that $f$ is invertible, as for most cases the factors are readily apparent form the image. We assume $f$ is smooth, because a small change in the factors should only result in a small change in the image and vice versa. We denote the inverse of the rendering engine as $f^{-1} = [f_{\mathbf{v}}^{-1}, f_{\mathbf{c}}^{-1}]$, where the subscript refers to the recovered factor.

**Weak labeling.** In the training we are given pairs of images $\mathbf{x}_1$ and $\mathbf{x}_2$, where they differ in $\mathbf{v}$ (varying factor), but they have the same $\mathbf{c}$ (common factor). We also assume that the two varying factors and the common factor are sampled independently, $\mathbf{v}_1 \sim p_{\mathbf{v}}$, $\mathbf{v}_2 \sim p_{\mathbf{v}}$ and $\mathbf{c} \sim p_{\mathbf{c}}$. The images are generated as $\mathbf{x}_1 = f(\mathbf{v}_1, \mathbf{c})$ and $\mathbf{x}_1 = f(\mathbf{v}_2, \mathbf{c})$. We call this labeling weak, because we do not know the absolute values of either the $\mathbf{v}$ or $\mathbf{c}$ factors or even relative changes between $\mathbf{v}_1$ and $\mathbf{v}_2$. All we know is that the image pairs share the same common factor.

**The encoder.** Let Enc be the encoder mapping images to features. For simplicity, we consider features split into only two column subvectors, $N_{\mathbf{v}}$ and $N_{\mathbf{c}}$, one associated to the varying factor $\mathbf{v}$ and the other associated to the common factor $\mathbf{c}$. Then, we have that $\text{Enc}(\mathbf{x}) = [N_{\mathbf{v}}(\mathbf{x}), N_{\mathbf{c}}(\mathbf{x})]$. Ideally, we would like to find the inverse of the image formation function, $[N_{\mathbf{v}}, N_{\mathbf{c}}] = f^{-1}$, which separates and recovers the factors $\mathbf{v}$ and $\mathbf{c}$ from data samples $\mathbf{x}$, *i.e.*,

$$N_{\mathbf{v}}(f(\mathbf{v}, \mathbf{c})) = \mathbf{v} \qquad N_{\mathbf{c}}(f(\mathbf{v}, \mathbf{c})) = \mathbf{c}. \tag{1}$$

In practice, this is not possible because any bijective transformation of $\mathbf{v}$ and $\mathbf{c}$ could be undone by $f$ and produce the same output $\mathbf{x}$. Therefore, we aim for $N_{\mathbf{v}}$ and $N_{\mathbf{c}}$ that satisfy the following *feature disentangling* properties

$$R_{\mathbf{v}}(N_{\mathbf{v}}(f(\mathbf{v}, \mathbf{c}))) = \mathbf{v} \qquad R_{\mathbf{c}}(N_{\mathbf{c}}(f(\mathbf{v}, \mathbf{c}))) = \mathbf{c} \tag{2}$$

for all $\mathbf{v}$, $\mathbf{c}$, and for some bijective functions $R_{\mathbf{v}}$ and $R_{\mathbf{c}}$, so that $N_{\mathbf{v}}$ is invariant to $\mathbf{c}$ and $N_{\mathbf{c}}$ is invariant to $\mathbf{v}$.

**The decoder.** Let Dec be the decoder mapping features to images. The sequence encoder-decoder is constrained to form an *autoencoder*, so

$$\text{Dec}(N_{\mathbf{v}}(\mathbf{x}), N_{\mathbf{c}}(\mathbf{x})) = \mathbf{x}, \qquad \forall \mathbf{x}. \tag{3}$$

To use the decoder for image synthesis, so that each input subvector affects only one factor in the rendered image, the ideal decoder should satisfy the *data disentangling* property

$$\text{Dec}(N_{\mathbf{v}}(f(\mathbf{v}_1, \mathbf{c}_1)), N_{\mathbf{c}}(f(\mathbf{v}_2, \mathbf{c}_2))) = f(\mathbf{v}_1, \mathbf{c}_2) \tag{4}$$

for any $\mathbf{v}_1$, $\mathbf{v}_2$, $\mathbf{c}_1$, and $\mathbf{c}_2$. The equation above describes the transfer of the varying factor $\mathbf{v}_1$ of $\mathbf{x}_1$ and the common factor $\mathbf{c}_2$ of $\mathbf{x}_2$ to a new image $\mathbf{x}_{1\oplus 2} = f(\mathbf{v}_1, \mathbf{c}_2)$.

In the next section we describe our training method for disentangling. We introduce a novel adversarial term, that does not need to be conditioned on the common factor, rather it uses only image pairs, that keeps the model parameters constant. Then we address the two main challenges of disentangling, the *shortcut problem* and the *reference ambiguity*. We discuss which disentanglement properties can be (provably) achieved by our (or any) method.

## 3.1 MODEL TRAINING

In our training procedure we use two terms in the objective function: an *autoencoder loss* and an *adversarial loss*. We describe these losses in functional form, however the components are implemented using neural networks. In all our terms we use the following sampling of independent factors

$$\mathbf{c}_1, \mathbf{c}_3 \sim p_{\mathbf{c}}, \quad \mathbf{v}_1, \mathbf{v}_2, \mathbf{v}_3 \sim p_{\mathbf{v}}. \tag{5}$$

The images are formed as $\mathbf{x}_1 = f(\mathbf{v}_1, \mathbf{c}_1)$, $\mathbf{x}_2 = f(\mathbf{v}_2, \mathbf{c}_1)$ and $\mathbf{x}_3 = f(\mathbf{v}_3, \mathbf{c}_3)$. The images $\mathbf{x}_1$ and $\mathbf{x}_2$ share the same common factor, and $\mathbf{x}_1$ and $\mathbf{x}_3$ are independent. In our objective functions, we use either pairs or triplets of the above images.

**Autoencoder loss.** In this term, we use images $\mathbf{x}_1$ and $\mathbf{x}_2$ with the same common factor $\mathbf{c}_1$. We feed both images to the encoder. Since both images share the same $\mathbf{c}_1$, we impose that the decoder should reconstruct $\mathbf{x}_1$ from the encoder subvector $N_{\mathbf{v}}(\mathbf{x}_1)$ and the encoder subvector $N_{\mathbf{c}}(\mathbf{x}_2)$, and similarly for the reconstruction of $\mathbf{x}_2$. The autoencoder objective is thus defined as

$$\mathcal{L}_{AE} \doteq E_{\mathbf{x}_1, \mathbf{x}_2}\Big[\big|\mathbf{x}_1 - \text{Dec}(N_{\mathbf{v}}(\mathbf{x}_1), N_{\mathbf{c}}(\mathbf{x}_2))\big|^2 + \big|\mathbf{x}_2 - \text{Dec}(N_{\mathbf{v}}(\mathbf{x}_2), N_{\mathbf{c}}(\mathbf{x}_1))\big|^2\Big]. \tag{6}$$

**Adversarial loss.** We introduce an adversarial training where the *generator* is our encoder-decoder pair and the *discriminator* Dsc is a neural network, which takes image pairs as input. The discriminator learns to distinguish between real image pairs $[\mathbf{x}_1, \mathbf{x}_2]$ and fake ones $[\mathbf{x}_1, \mathbf{x}_{3\oplus 1}]$, where $\mathbf{x}_{3\oplus 1} \doteq \text{Dec}(N_{\mathbf{v}}(\mathbf{x}_3), N_{\mathbf{c}}(\mathbf{x}_1))$. If the encoder were ideal, the image $\mathbf{x}_{3\oplus 1}$ would be the result of taking the common factor from $\mathbf{x}_1$ and the varying factor from $\mathbf{x}_3$. The generator learns to fool the discriminator, so that $\mathbf{x}_{3\oplus 1}$ looks like the random variable $\mathbf{x}_2$ (the common factor is $\mathbf{c}_1$ and the varying factor is independent of $\mathbf{v}_1$). To this purpose, the decoder must make use of $N_{\mathbf{c}}(\mathbf{x}_1)$, since $\mathbf{x}_3$ does not carry any information about $\mathbf{c}_1$. The objective function is thus defined as

$$\mathcal{L}_{GAN} \doteq E_{\mathbf{x}_1, \mathbf{x}_2}\Big[\log(\text{Dsc}(\mathbf{x}_1, \mathbf{x}_2))\Big] + E_{\mathbf{x}_1, \mathbf{x}_3}\Big[\log(1 - \text{Dsc}(\mathbf{x}_1, \mathbf{x}_{3\oplus 1}))\Big]. \tag{7}$$

**Composite loss.** Finally, we optimize the weighted sum of the two losses $\mathcal{L} = \mathcal{L}_{AE} + \lambda\mathcal{L}_{GAN}$,

$$\min_{\text{Dec,Enc}} \max_{\text{Dsc}} \mathcal{L}_{AE}(\text{Dec}, \text{Enc}) + \lambda\mathcal{L}_{GAN}(\text{Dec}, \text{Enc}, \text{Dsc}) \tag{8}$$

where $\lambda$ regulates the relative importance of the two losses.

## 3.2 SHORTCUT PROBLEM.

Ideally, at the global minimum of $\mathcal{L}_{AE}$, $N_{\mathbf{v}}$ relates only to the factor $\mathbf{v}$ and $N_{\mathbf{c}}$ only to $\mathbf{c}$. However, the encoder may map a complete description of its input into $N_{\mathbf{v}}$ and the decoder may completely ignore $N_{\mathbf{c}}$. We call this challenge the *shortcut problem*. When the shortcut problem occurs, the decoder is invariant to its second input, so it does not transfer the $\mathbf{c}$ factor correctly,

$$\text{Dec}(N_{\mathbf{v}}(\mathbf{x}_3), N_{\mathbf{c}}(\mathbf{x}_1)) = \mathbf{x}_3. \tag{9}$$

The shortcut problem can be addressed by reducing the dimensionality of $N_\mathbf{v}$, so it cannot build a complete representation of all input images. This also forces the encoder and decoder to make use of $N_\mathbf{c}$ for the common factor. However, this strategy may not be convenient as it leads to a time consuming trial-and-error procedure to find the correct dimensionality. A better way to address the shortcut problem is to use adversarial training (7) (8).

**Proposition 1.** *Let $\mathbf{x}_1$, $\mathbf{x}_2$ and $\mathbf{x}_3$ data samples generated according to* (5), *where the factors $\mathbf{c}_1, \mathbf{c}_3, \mathbf{v}_1, \mathbf{v}_2, \mathbf{v}_3$ are jointly independent, and $\mathbf{x}_{3\oplus1} \doteq Dec(N_\mathbf{v}(\mathbf{x}_3), N_\mathbf{c}(\mathbf{x}_1))$. When the global optimum of the composite loss* (8) *is reached, the $\mathbf{c}$ factor is transferred to $\mathbf{x}_{3\oplus1}$, i.e. $f_\mathbf{c}^{-1}(\mathbf{x}_{3\oplus1}) = \mathbf{c}_1$.*

*Proof.* When the global optimum of (8) is reached, the distribution of real $[\mathbf{x}_1, \mathbf{x}_2]$ and fake $[\mathbf{x}_1, \mathbf{x}_{3\oplus1}]$ image pairs are identical. We compute statistics of the inverse of the rendering engine of the common factor $f_\mathbf{c}^{-1}$ on the data. For the images $\mathbf{x}_1$ and $\mathbf{x}_2$ we obtain

$$E_{\mathbf{x}_1,\mathbf{x}_2}\left[|f_\mathbf{c}^{-1}(\mathbf{x}_1) - f_\mathbf{c}^{-1}(\mathbf{x}_2)|^2\right] = E_{\mathbf{c}_1}\left[|\mathbf{c}_1 - \mathbf{c}_1|^2\right] = 0 \tag{10}$$

by construction (of $\mathbf{x}_1$ and $\mathbf{x}_2$). For the images $\mathbf{x}_1$ and $\mathbf{x}_{3\oplus1}$ we obtain

$$E_{\mathbf{x}_1,\mathbf{x}_3}\left[|f_\mathbf{c}^{-1}(\mathbf{x}_1) - f_\mathbf{c}^{-1}(\mathbf{x}_{3\oplus1})|^2\right] = E_{\mathbf{v}_1,\mathbf{c}_1,\mathbf{v}_3,\mathbf{c}_3}\left[|\mathbf{c}_1 - \mathbf{c}_{3\oplus1}|^2\right] \geq 0, \tag{11}$$

where $\mathbf{c}_{3\oplus1} = f_\mathbf{c}^{-1}(\mathbf{x}_{3\oplus1})$. We achieve equality if and only if $\mathbf{c}_1 = \mathbf{c}_{3\oplus1}$ everywhere. $\qquad\square$

### 3.3 REFERENCE AMBIGUITY

Let us consider the ideal case where we observe the space of all images. When weak labels are made available to us, we also know what images $\mathbf{x}_1$ and $\mathbf{x}_2$ share the same $\mathbf{c}$ factor (for example, which images have the same car). This labeling is equivalent to defining the probability density function $p_\mathbf{c}$ and the joint conditional $p_{\mathbf{x}_1,\mathbf{x}_2|\mathbf{c}}$, where

$$p_{\mathbf{x}_1,\mathbf{x}_2|\mathbf{c}}(\mathbf{x}_1, \mathbf{x}_2|\mathbf{c}) = \int \delta(\mathbf{x}_1 - f(\mathbf{v}_1, \mathbf{c}))\delta(\mathbf{x}_2 - f(\mathbf{v}_2, \mathbf{c}))p(\mathbf{v}_1)p(\mathbf{v}_2)d\mathbf{v}_1 d\mathbf{v}_2. \tag{12}$$

Firstly, we show that the labeling allows us to satisfy the feature disentangling property for $\mathbf{c}$ (2). For any $[\mathbf{x}_1, \mathbf{x}_2] \sim p_{\mathbf{x}_1,\mathbf{x}_2|\mathbf{c}}$ we impose $N_\mathbf{c}(\mathbf{x}_1) = N_\mathbf{c}(\mathbf{x}_2)$. In particular, this equation is true for pairs when one of the two images is held fixed. Thus, a function $C(\mathbf{c}) = N_\mathbf{c}(\mathbf{x}_1)$ can be defined, where the $C$ only depends on $\mathbf{c}$, because $N_\mathbf{c}$ is invariant to $\mathbf{v}$. Lastly, images with the same $\mathbf{v}$, but different $\mathbf{c}$ must also result in different features, $C(\mathbf{c}_1) = N_\mathbf{v}(f(\mathbf{v}, \mathbf{c}_1)) \neq N_\mathbf{v}(\mathbf{v}, \mathbf{c}_2) = C(\mathbf{c}_2)$, otherwise the autoencoder constraint (3) cannot be satisfied. Then, there exists a bijective function $R_\mathbf{c} = C^{-1}$ such that property (2) is satisfied for $\mathbf{c}$. Unfortunately the other disentangling properties can not provably be satisfied.

**Definition 1.** *A function $g$ reproduces the data distribution, when it generates samples $\mathbf{y}_1 = g(\mathbf{v}_1, \mathbf{c})$ and $\mathbf{y}_2 = g(\mathbf{v}_2, \mathbf{c})$ that have the same distribution as the data. Formally, $[\mathbf{y}_1, \mathbf{y}_2] \sim p_{\mathbf{x}_1,\mathbf{x}_2}$, where the latent factors are independent, $\mathbf{v}_1 \sim p_\mathbf{v}$, $\mathbf{v}_2 \sim p_\mathbf{v}$ and $\mathbf{c} \sim p_\mathbf{c}$.*

The reference ambiguity occurs, when a decoder reproduces the data without satisfying the disentangling properties.

**Proposition 2.** *Let $p_\mathbf{v}$ assign the same probability value to at least two different instances of $\mathbf{v}$. Then, we can find encoders that reproduce the data distribution, but do not satisfy the disentangling properties for $\mathbf{v}$ in* (2) *and* (4).

*Proof.* We already saw that $N_\mathbf{c}$ satisfies (2), so we can choose $N_\mathbf{c} = f_\mathbf{c}^{-1}$, the inverse of the rendering engine. Now we look at defining $N_\mathbf{v}$ and the decoder. The iso-probability property of $p_\mathbf{v}$ implies that there exists a mapping $T(\mathbf{v}, \mathbf{c})$, such that $T(\mathbf{v}, \mathbf{c}) \sim p_\mathbf{v}$ and $T(\mathbf{v}, \mathbf{c}_1) \neq T(\mathbf{v}, \mathbf{c}_2)$ for some $\mathbf{v}$ and $\mathbf{c}_1 \neq \mathbf{c}_2$. For example, let us denote with $\mathbf{v}_1 \neq \mathbf{v}_2$ two varying components such that $p_\mathbf{v}(\mathbf{v}_1) = p_\mathbf{v}(\mathbf{v}_2)$. Then, let

$$T(\mathbf{v}, \mathbf{c}) \doteq \begin{cases} \mathbf{v} & \text{if } \mathbf{v} \neq \mathbf{v}_1, \mathbf{v}_2 \\ \mathbf{v}_1 & \text{if } \mathbf{v} = \mathbf{v}_1 \vee \mathbf{v}_2 \text{ and } \mathbf{c} \in \mathcal{C} \\ \mathbf{v}_2 & \text{if } \mathbf{v} = \mathbf{v}_1 \vee \mathbf{v}_2 \text{ and } \mathbf{c} \notin \mathcal{C} \end{cases} \tag{13}$$

and $\mathcal{C}$ is a subset of the domain of $\mathbf{c}$, where $\int_{\mathcal{C}} p_\mathbf{c}(\mathbf{c})d\mathbf{c} = 1/2$. Now, let us define the encoder as $N_\mathbf{v}(f(\mathbf{v}, \mathbf{c})) = T(\mathbf{v}, \mathbf{c})$. By using the autoencoder constraint, the decoder satisfies

$$\text{Dec}(N_\mathbf{v}(f(\mathbf{v}, \mathbf{c})), N_\mathbf{c}(f(\mathbf{v}, \mathbf{c}))) = \text{Dec}(T(\mathbf{v}, \mathbf{c}), \mathbf{c}) = f(\mathbf{v}, \mathbf{c}). \quad (14)$$

Even though $T(\mathbf{v}, \mathbf{c})$ depends on $\mathbf{c}$ functionally, they are statistically independent. Because $T(\mathbf{v}, \mathbf{c}) \sim p_\mathbf{v}$ and $\mathbf{c} \sim p_\mathbf{c}$ by construction, our encoder-decoder pair defines a data distribution identical to that given as training set

$$[\text{Dec}(T(\mathbf{v}_1, \mathbf{c}), \mathbf{c}), \text{Dec}(T(\mathbf{v}_2, \mathbf{c}), \mathbf{c})] \sim p_{\mathbf{x}_1, \mathbf{x}_2}. \quad (15)$$

The feature disentanglement property is not satisfied because $N_\mathbf{v}(f(\mathbf{v}_1, \mathbf{c}_1)) = T(\mathbf{v}_1, \mathbf{c}_1) \neq T(\mathbf{v}_1, \mathbf{c}_2) = N_\mathbf{v}(f(\mathbf{v}_1, \mathbf{c}_2))$, when $\mathbf{c}_1 \in \mathcal{C}$ and $\mathbf{c}_2 \notin \mathcal{C}$. Similarly, the data disentanglement property does not hold, because $\text{Dec}(T(\mathbf{v}_1, \mathbf{c}_1), \mathbf{c}_1) \neq \text{Dec}(T(\mathbf{v}_1, \mathbf{c}_2), \mathbf{c}_2)$. $\qquad\square$

The above proposition implies that we cannot provably disentangle all the factors of variation from weakly labeled data, even if we had access to all the data and knew the distributions $p_\mathbf{v}$ and $p_\mathbf{c}$.

To better understand it, let us consider a practical example. Let $\mathbf{v} \sim \mathcal{U}[-\pi, \pi]$ be the (continuous) viewpoint (the azimuth angle) and $\mathbf{c} \sim \mathcal{B}(0.5)$ the car type, where $\mathcal{U}$ denotes the uniform distribution and $\mathcal{B}(0.5)$ the Bernoulli distribution with probability $p_\mathbf{c}(\mathbf{c} = 0) = p_\mathbf{c}(\mathbf{c} = 1) = 0.5$ (*i.e.*, there are only 2 car types). In this case, every instance of $\mathbf{v}$ is iso-probable in $p_\mathbf{v}$ so we have the worst scenario for the reference ambiguity. We can define the function $T(\mathbf{v}, \mathbf{c}) = \mathbf{v}(2\mathbf{c} - 1)$ so that the mapping of $\mathbf{v}$ is mirrored as we change the car type. By construction $T(\mathbf{v}, \mathbf{c}) \sim \mathcal{U}[-\pi, \pi]$ for any $\mathbf{c}$ and $T(\mathbf{v}, \mathbf{c}_1) \neq T(\mathbf{v}, \mathbf{c}_2)$ for $\mathbf{v} \neq 0$ and $\mathbf{c}_1 \neq \mathbf{c}_2$. So we cannot tell the difference between $T$ and the ideal correct mapping to the viewpoint factor. This is equivalent to an encoder $N_\mathbf{v}(f(\mathbf{v}, \mathbf{c})) = T(\mathbf{v}, \mathbf{c})$ that reverses the ordering of the azimuth of car 1 with respect to car 0. Each car has its own reference system, and thus it is not possible to transfer the viewpoint from one system to the other, as it is illustrated in fig. 1.

## 3.4 Implementation

In our implementation we use convolutional neural networks for all the models. We denote with $\theta$ the parameters associated to each network. Then, the optimization of the composite loss can be written as

$$\hat\theta_\text{Dec}, \hat\theta_\text{Enc}, \hat\theta_\text{Dsc} = \arg \min_{\theta_\text{Dec}, \theta_\text{Enc}} \max_{\theta_\text{Dsc}} \mathcal{L}(\theta_\text{Dec}, \theta_\text{Enc}, \theta_\text{Dsc}). \quad (16)$$

We choose $\lambda = 1$ and also add regularization to the adversarial loss so that each logarithm has a minimum value. We define $\log_\epsilon \text{Dsc}(\mathbf{x}_1, \mathbf{x}_2) = \log(\epsilon + \text{Dsc}(\mathbf{x}_1, \mathbf{x}_2))$ (and similarly for the other logarithmic term) and use $\epsilon = 10^{-12}$. The main components of our neural network are shown in Fig. 2. The architecture of the encoder and the decoder were taken from DCGAN Radford et al. (2015), with slight modifications. We added fully connected layers at the output of the encoder and to the input of the decoder. For the discriminator we used a simplified version of the VGG Simonyan & Zisserman (2014) network. As the input to the discriminator is an image pair, we concatenate them along the color channels.

**Normalization.** In our architecture both the encoder and the decoder networks use blocks with a convolutional layer, a nonlinear activation function (ReLU/leaky ReLU) and a normalization layer, typically, batch normalization (BN). As an alternative to BN we consider the recently introduced *instance normalization* (IN) Ulyanov et al. (2017). The main difference between BN and IN is that the latter just computes the mean and standard deviation across the spatial domain of the input and not along the batch dimension. Thus, the shift and scaling for the output of each layer is the same at every iteration for the same input image. In practice, we find that IN improves the performance.

## 4 Experiments

We tested our method on the MNIST, Sprites and ShapeNet datasets. We performed ablation studies on the shortcut problem using ShapeNet cars. We focused on the effect of the feature dimensionality and having the adversarial term ($\mathcal{L}_{AE} + \mathcal{L}_{GAN}$) or not ($\mathcal{L}_{AE}$). We also show that in most cases the reference ambiguity does not arise in practice (MNIST, Sprites, ShapeNet cars), we can only observe it when the data is more complex (ShapeNet chairs).

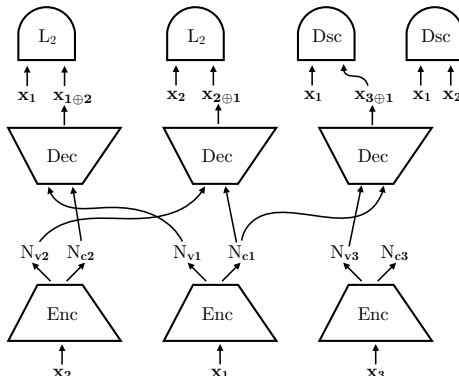

Figure 2: Learning to disentangle factors of variation. The scheme above shows how the encoder (Enc), the decoder (Dec) and the discriminator (Dsc) are trained with input triplets. The components with the same name share weights.

## 4.1 SHORTCUT PROBLEM

**ShapeNet cars.** The ShapeNet dataset Chang et al. (2015) contains 3D objects than we can render from different viewpoints. We consider only one category (cars) for a set of fixed viewpoints. Cars have high intraclass variability and they do not have rotational symmetries. We used approximately 3K car types for training and 300 for testing. We rendered 24 possible viewpoints around each object in a full circle, resulting in 80K images in total. The elevation was fixed to 15 degrees and azimuth angles were spaced 15 degrees apart. We normalized the size of the objects to fit in a $100 \times 100$ pixel bounding box, and placed it in the middle of a $128 \times 128$ pixel image.

Fig. 3 shows the attribute transfer on the Shapenet cars. We compare the methods $\mathcal{L}_{AE}$ and $\mathcal{L}_{AE} + \mathcal{L}_{GAN}$ with different feature dimension of $N_\mathbf{v}$. The size of the common feature $N_\mathbf{c}$ was fixed to 1024 dimensions. We can observe that the transferring performance degrades for $\mathcal{L}_{AE}$, when we increase the feature size of $N_\mathbf{v}$. As expected, the autoencoder takes the shortcut and tries to store all information into $N_\mathbf{v}$. The model $\mathcal{L}_{AE} + \mathcal{L}_{GAN}$ instead renders images without loss of quality, independently of the feature dimension.

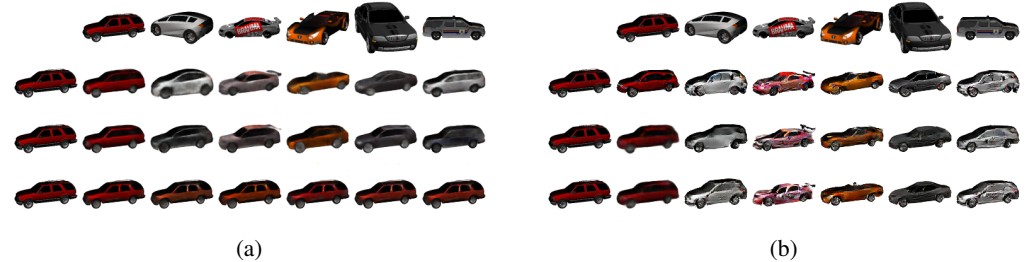

(a)  (b)

Figure 3: Feature transfer on Shapenet. (a) synthesized images with $\mathcal{L}_{AE}$, where the top row shows images from which the car type is taken. The second, third and fourth row show the decoder renderings using 2, 16 and 128 dimensions for the feature $N_\mathbf{v}$. (b) images synthesized with $\mathcal{L}_{AE} + \mathcal{L}_{GAN}$. The setting for the inputs and feature dimensions are the same as in (a).

In Fig. 4 we visualize the t-SNE embeddings of the $N_\mathbf{v}$ features for several models using different feature sizes. For the $2D$ case, we do not modify the data. We can see that both $\mathcal{L}_{AE}$ with 2 dimensions and $\mathcal{L}_{AE} + \mathcal{L}_{GAN}$ with 128 separate the viewpoints well, but $\mathcal{L}_{AE}$ with 128 dimensions does not due to the shortcut problem. We investigate the effect of dimensionality of the $N_\mathbf{v}$ features on the nearest neighbor classification task. The performance is measured by the mean average precision. For $N_\mathbf{v}$ we use the viewpoint as ground truth. Fig. 4 also shows the results on $\mathcal{L}_{AE}$ and $\mathcal{L}_{AE} + \mathcal{L}_{GAN}$ models with different $N_\mathbf{v}$ feature dimensions. The dimension of $N_\mathbf{c}$ was fixed to 1024

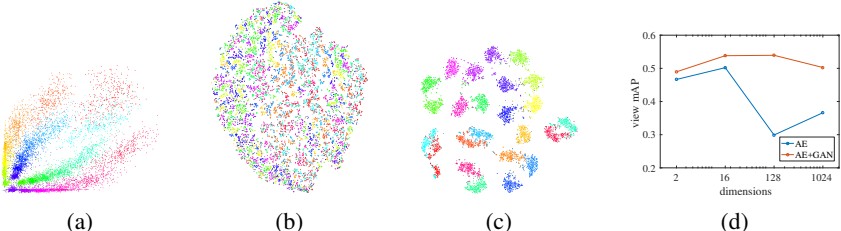

(a)  (b)  (c)  (d)

Figure 4: The effect of dimensions and objective function on $N_v$ features. (a), (b), (c) t-SNE embeddings on $N_{\mathbf{v}}$ features. Colors correspond to the ground truth viewpoint. The objective functions and the $N_{\mathbf{v}}$ dimensions are: (a) $\mathcal{L}_{AE}$ 2 dim, (b) $\mathcal{L}_{AE}$ 128 dim, (c) $\mathcal{L}_{AE} + \mathcal{L}_{GAN}$ 128 dim. (d) Mean average precision curves for the viewpoint prediction from the viewpoint feature using different models and dimensions for $N_{\mathbf{v}}$.

Table 1: Nearest neighbor classification on $N_{\mathbf{v}}$ and $N_{\mathbf{c}}$ features using different normalization techniques on ShapeNet cars.

| Normalization | $N_{\mathbf{v}}$ mAP | $N_{\mathbf{c}}$ mAP |
|---|---|---|
| **None** | 0.47 | 0.13 |
| **Batch** | 0.50 | 0.08 |
| **Instance** | 0.50 | 0.20 |

for this experiment. One can now see quantitatively that $\mathcal{L}_{AE}$ is sensitive to the size of $N_{\mathbf{v}}$, while $\mathcal{L}_{AE} + \mathcal{L}_{GAN}$ is not. $\mathcal{L}_{AE} + \mathcal{L}_{GAN}$ also achieves a better performance.

We compare the different normalization choices in Table 1. We evaluate the case when batch, instance and no normalization are used and compute the performance on the nearest neighbor classification task. We fixed the feature dimensions at 1024 for both $N_{\mathbf{v}}$ and $N_{\mathbf{c}}$ features in all normalization cases. We can see that both batch and instance normalization perform equally well on viewpoint classification and no normalization is slightly worse. For the car type classification instance normalization is clearly better.

## 4.2 REFERENCE AMBIGUITY

**MNIST.** The MNIST dataset LeCun et al. (1998) contains handwritten grayscale digits of size $28 \times 28$ pixel. There are 60K images of 10 classes for training and 10K for testing. The common factor is the digit class and the varying factor is the intraclass variation. We take image pairs that have the same digit for training, and use our full model $\mathcal{L}_{AE} + \mathcal{L}_{GAN}$ with dimensions 64 for $N_{\mathbf{v}}$ and 64 for $N_{\mathbf{c}}$. In Fig. 5 (a) and (b) we show the transfer of varying factors. Qualitatively, both our method and Mathieu et al. (2016) perform well. We observe neither the reference ambiguity nor the shortcut problem in this case.

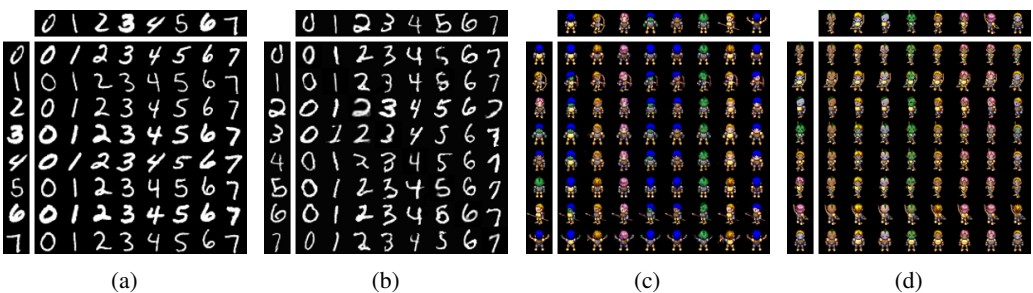

(a)  (b)  (c)  (d)

Figure 5: Renderings of transferred features. In all figures the variable factor is transferred from the left column and the common factor from the top row. (a) MNIST Mathieu et al. (2016); (b) MNIST (ours); (c) Sprites Mathieu et al. (2016); (d) Sprites (ours).

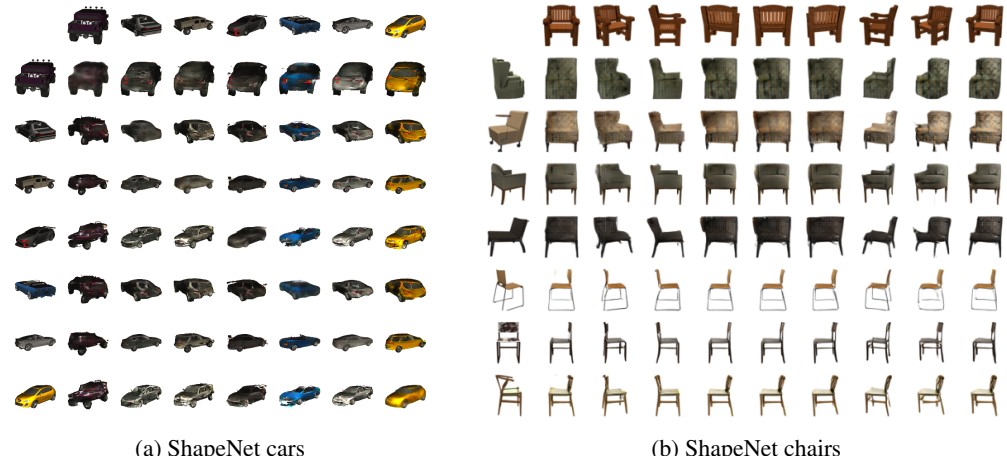

(a) ShapeNet cars            (b) ShapeNet chairs

Figure 6: Attribute transfer on ShapeNet. For both subfigures the viewpoint is taken from the leftmost column and the car/chair type is taken from the first row. (a) Cars: the factors are transferred correctly. (b) Chairs: in the bottom three rows the viewpoint is not transferred correctly due to the reference ambiguity.

**Sprites.** The Sprites dataset Reed et al. (2015) contains 60 pixel color images of animated characters (sprites). There are 672 sprites, 500 for training, 100 for testing and 72 for validation. Each sprite has 20 animations and 178 images, so the full dataset has 120K images in total. There are many changes in the appearance of the sprites, they differ in their body shape, gender, hair, armour, arm type, greaves, and weapon. We consider character identity as the common factor and the pose as the varying factor. We train our system using image pairs of the same sprite and do not exploit labels on their pose. We train the $\mathcal{L}_{AE} + \mathcal{L}_{GAN}$ model with dimensions 64 for $N_\mathbf{v}$ and 448 for $N_\mathbf{c}$. Fig. 5 (c) and (d) show results on the attribute transfer task. Both our method and Mathieu et al. (2016)'s transfer the identity of the sprites correctly, the reference ambiguity does not arise.

**ShapeNet chairs.** We render the ShapeNet chairs with the same settings (viewpoints, image size) as the cars. There are 3500 chair types for training and 3200 for testing, so the dataset contains 160K images. We trained $\mathcal{L}_{AE} + \mathcal{L}_{GAN}$, and set the feature dimensions to 1024 for both $N_\mathbf{v}$ and $N_\mathbf{c}$. In Fig. 6 we show results on attribute transfer and compare it with ShapeNet cars. We found that the reference ambiguity does not emerge for cars, but it does for chairs, possibly due to the higher complexity, as cars have much less variability than chairs.

## 5   CONCLUSIONS

In this paper we studied the challenges of disentangling factors of variation, mainly the shortcut problem and the reference ambiguity. The shortcut problem occurs when all information is stored in only one feature chunk, while the other is ignored. The reference ambiguity means that the reference in which a factor is interpreted, may depend on other factors. This makes the attribute transfer ambiguous. We introduced a novel training of autoencoders to solve disentangling using image triplets. We showed theoretically and experimentally how to keep the shortcut problem under control through adversarial training, and enable to use large feature dimensions. We proved that the reference ambiguity is inherently present in the disentangling task when weak labels are used. Most importantly this can be stated independently of the learning algorithm. We demonstrated that training and transfer of factors of variation may not be guaranteed. However, in practice we observe that our trained model works well on many datasets and exhibits good generalization capabilities.

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
