# OpenReview forum: "Challenges in Disentangling Independent Factors of Variation"
_ICLR.cc/2018/Conference — Invite to Workshop Track_

### Official Review · AnonReviewer2 · 2017-11-23
**Progress on disentangling; could be written more clearly.**

**Rating:** 6
**Confidence:** 4

**Review:**

Quality
The method description, particularly about reference ambiguity, I found difficult to follow. The experiments and analysis look solid, although it would be nice to see experiments on more challenging natural image datasets.

Clarity
“In general this is not possible… “ - you are saying it is not possible to learn an encoder that recovers disentangled factors of variation? But that seems to be one of the main goals of the paper. It is not clear at all what is meant here or what the key problem is, which detracts from the paper’s motivation.

What is the purpose of R_v and R_c in eq 2? Why can these not be collapsed into the encoders N_v and N_c?

What does “different common factor” mean?

What is f_c in proof of proposition 1? Previously f (no subscript) was referred to as a rendering engine.

T(v,c) ~ p_v and c ~ p_c are said to be independent. But T(v,c) is explicitly defined in terms of c (equation 6). So which is correct?

Overall the argument seems plausible - pairs of images in which a single factor of variation changes have a reference ambiguity - but the details are unclear.

Originality

The model is very similar to Mathieu et al, although using image pairs rather than category labels directly. The idea of weakly-supervised disentangling has also been explored in many other papers, e.g. “Weakly-supervised Disentangling with Recurrent Transformations for 3D View Synthesis”, Yang et al. The description of reference ambiguity seems new and potentially valuable, but I did not find it easy to follow.

Significance

Disentangling factors of variation with weak supervision is an important problem, and this paper makes a modest advance in terms of the model and potentially in terms of the theory. The analysis in figure 3 I found particularly interesting - illustrating that the encoder embedding dimension can have a drastic effect on the shortcut problem. Overall I think this can be a significant contribution if the exposition can be improved.

Pros
- Proposed method allows disentangling two factors of variation given a training set of image pairs with one factor of variation matching and the other non-matching.
- A challenge inherent to weakly supervised disentangling called reference ambiguity is described.

Cons
- Only two factors of variation are studied, and the datasets are fairly simple.
- The method description and the description of reference ambiguity are unclear.

---

> ### Author Response · Authors · 2017-12-13
> **Comments on clarity and significance, and clarification of details**
>
> Thank you for your feedback.
>
> The reference ambiguity is an inherent ambiguity in the disentanglement task itself. So there is no algorithm that provably solves it. The emphasis here is on the "provably" part. In practice most methods work on most datasets. The question is why? This is a very similar question to why neural networks learn useful features in an autoencoder or why the features are transferrable to learning tasks other than they were trained on.
>
> Our work is a step towards understanding this problem, as it highlights that we cannot do that by reasoning in terms of attribute/feature distributions. As a consequence when the reference ambiguity arises, a better GAN objective will most likely not solve the issue.
>
> Specific questions:
> 1. N_v and N_c refers to possible trained encoders. R_v and R_c is meant to create an equivalence relation between the possible encoders.
> 2. "different common factor" meant different c.
> 3. f_c^-1 is the inverse of the rendering engine for the c factor. We revised the proof to make it clearer.
> 4. T(v, c) functionally depends on c, but it is designed in a way that they are statistically independent.

---

### Official Review · AnonReviewer1 · 2017-11-26
**The paper offers some interesting ideas. However, the presentation is somewhat confusing and the resulting architecture does not seem justified by the theory**

**Rating:** 5
**Confidence:** 3

**Review:**

The paper considers the challenges of disentangling factors of variation in images: for example disentangling viewpoint from vehicle type in an image of a car. They identify a well-known problem, which they call "reference ambiguity", and show that in general without further assumptions one cannot tell apart two different factors of variation.

They then go on to suggest an interesting AE+GAN architecture where the main novelty is the idea of taking triplets such that the first two instances vary in only one factor of variation, while the third instance varies in both from the pair. This is clever and allows them to try and disentangle the variation factors using a joint encoder-decoder architecture working on the triplet.

Pros:
1. Interesting use of constructed triplets.
2. Interesting use of GAN on the artificial instance named x_{3 \oplus 1}

Cons:
1. Lack of clarity: the paper is hard to follow at times. It's not entirely obvious how the theoretical part informs the practical part. See detailed comments below.
2. The theory addresses two widely recognized problems as if they're novel:  "reference ambiguity" and "shortcut problem". The second merely refers to the fact that unconstrained autoencoders will merely memorize the instance.
3. Some of the architectural choices (the one derived from "shortcut problem") are barely explained or looked into.

Specific comments:

1. An important point regarding the reference ambiguity problem and eq. (2): a general bijective function mixing v and c would not have the two components as independent. The authors could have used this extremely important aspect of the generative process they posit in order to circumvent the problem of ambiguity. In fact, I suspect that this is what allows their method to succeed.

2. I think the intro could be made better if more concrete examples be made earlier on. Specifically the car-type/viewpoint example, along with noting what weak labels mean in that context.

3. In presenting autoencoders it is crucial to note that they are all built around the idea of compression. Otherwise, the perfect latent representation is z=x.

4. I would consider switching the order of sections 2 and 3, so the reader will be better grounded in what this paper is about before reading the related work.

5. In discussing attributes and "valid" features, I found the paper rather vague. An image has many attributes: the glint in the corner of a window, the hue of a leaf. The authors should be much more specific in this discussion and definite explicitly and clearly what they mean when they use these terms.

6. In equation (5), should it be p(v_1,v_2)? Or are v_1 and v_2 assumed to be independent?

7. Under equation (5), the paper mentions an "autoencoder constraint". Such a constraint is not mentioned up to this point in the paper if I'm not mistaken.

8. Also under equation (5): is this where the encoder requirements are defined? If so, please be more explicit about it. Also note that you should require c_1 \neq c_2.

9. In proof of Proposition 1, there is discussion of N_c. N_c was mentioned before but never properly defined; same for R_c and C^-1. These should be part of the proposition statement or defined formally. Currently they are only discussed ad-hoc after equation (5).

10 .In the proof of Proposition 1, what is f_c^-1 ? It's only defined later in the paper.

11. In general, what promises that f_c^-1 and f_v^-1 are well defined? Are f_c and f_v injective? Why?

12. Before explaining the training of the model, the task should be defined properly. What is the goal of the training?

13. In eq. (15) I am missing a term which addresses "the shortcut problem" as defined in the previous page.

14. The weak labels are never properly defined and are discussed in a vague manner. Please define what does that term mean in your context and what were the weak labels in each experiment.

15. In the conclusion, I would edit to say the "our trained model works well on *several* datasets".


Minor comments:
Please use \citep when appropriate. Instead of "Generative Adversarial Nets Goodfellow et al. (2014)", you should have "Generative Adversarial Nets (Goodfellow et al., 2014)"

---

> ### Author Response · Authors · 2017-12-13
> **Clarification of details**
>
> Thank you for your feedback. First we would like to address your main complaints of the paper:
>
> 1. lack of clarity:
> As multiple reviewers pointed out, the paper needs improvement in the presentation. We revised the paper accordingly, and we hope we could make it easier to understand.
> 2. reference ambiguity and shortcut problem
> As far as we know the description of the reference ambiguity for the weakly labelled case of the disentanglement problem is novel. If it is well known, please provide a reference.
> The shortcut problem occurs, when only one feature chunk contains the information about the input, and the decoder ignores the other chunk. Memorisation does not play a role in this.
> 3. architecture not explained
> We did not provide a lot of details of encoder/decoder/discriminator components, because we borrowed them from referred works, and that was not our main focus. We did specify however the most important architectural choice regarding the shortcut problem, namely the feature size. Moreover we did ablation studies on that.
>
> Specific comments:
> 1. In general N_v(f(v,c)) and N_c(f(v,c)) are not independent. When N_v and N_c are invariant to c and v respectively, they are independent. The reference ambiguity states the reverse is not true. We can find N_v and N_c that are independent, but N_v is still not invariant to c. Therefore training them to be independent is not enough. And training N_v to be invariant to c is not possible because of the lack of labels.
> 2. Thank you for this suggestion. We revised the introduction to visualise the attribute transfer along with the challenges: reference ambiguity and shortcut problem. We also clarify what the weak labelling means early on.
> 3. The shortcut problem is related to the compression indeed. Using higher compression and low dimensional features, the shortcut problem can be avoided. The role adversarial term is to make sure the disentanglement works regardless of the feature size.
> 4. We revised the intro instead.
> 5. Valid feature meant that it is part of the decoders domain, i.e. computed by the encoder from an image. In the revised paper we clarified these terms.
> 6. Yes, the image pairs have independent v attributes (viewpoints), this is our model assumption.
> 7. The autoencoder constraint is equation 3.
> 8. We show here that the weak labelling determines N_c up to a bijection.
> 9. N_c is the encoder (defined before equation 1), R_c is a bijection (defined in equation 2) and C is defined as C(c) = N_c(f(v,c)).
> 10. Thank you for noticing this, we fixed in the revised paper.
> 11. Our model assumption is that f is smooth and invertible. Smooth because a small change in the attribute should change the image a small amount and vice versa. We also assume the attributes are readily apparent from the image, hence f is invertible.
> 12. The goals of disentanglement are to achieve the feature and image/data level disentanglement (equation 2 and 4). The goal of the paper is to study how/if we can achieve them (provably).
> 13. The composite loss includes both AE and GAN terms. The GAN term addresses the shortcut problem.
> 14. We revised paper to better explain weak labels.

---

### Official Review · AnonReviewer3 · 2017-11-27
**What is the overarching goal? There is lack of clarity between the theory parts and the experiments.**

**Rating:** 5
**Confidence:** 3

**Review:**

This paper studies the challenges of disentangling independent factors of variation under weakly labeled data.

A term "reference ambiguity" is introduced, which refers to the fact that there is no guarantee that two data points with same factor of variation will be mapped to the same point if there is only weakly labeled data to that extend.

I am having a hard time understanding the message of the paper. The proof in section 3.1, although elementary, is nice. But then the authors train fairly standard networks in experiments (section 4) for datasets studied with these methods in the literature, and they fail to draw any connection to the introduced reference ambiguity concept.

As written, the paper to me looks like two separate white papers:
{beginning - to -end of section 3}: as a theoretical white paper that lacks experiments, and
{section 4}: experiments with some recent methods / datasets (this part is almost like a cute course project).

Either the paper lacks to harmonically present the over arching goal, or I have missed if there was any such message that was implicit in between the lines. A rewrite with strong connection between the theory and the experiments is required.

---

> ### Author Response · Authors · 2017-12-13
> **Comments on the paper's goals**
>
> Thank you for your feedback.
>
> The goal of the paper is to describe the challenges of disentangling factors of variation:
> - reference ambiguity: inherently present in the task
> - shortcut problem: specific to the swapping auto-encoder setting
> - we introduce a novel method for disentangling
>
> Our method has the advantage over previous methods (Mathieu etal.), that it does not need the common factor labels as inputs, keeping the trainable parameters constant. This is arguably an incremental improvement, but definitely novel and not a recent (existing) method. We also prove that our method solves the shortcut problem.
>
> In the experiments we show:
> - detailed ablation studies on the shortcut problem
> - in practice the reference ambiguity only appears on a complex dataset and not on the simpler ones
>
> We revised the paper in order to:
> - better highlight the shortcut problem and the related proofs and experiments
> - better experiment on the effects of the reference ambiguity

---

### Decision · Program_Chairs · 2018-01-29
**ICLR 2018 Conference Acceptance Decision**

**Decision:**

Invite to Workshop Track

**Comment:**

The paper proposes a method to disentangle style from content (two factor disentanglement) using weak labels (information about the common factor for a pair of images). It is similar to an earlier work by Mathieu et al (2016) with main novelty being in the use of the discriminator which operates with pairs of images in the proposed method. Authors also have some theoretical statements about two challenges in disentangling the factors but reviewers have complained about missing connection b/w theory and experiments, and about exposition in general.

The idea has novelty, although somewhat limited in the light of earlier work by Mathieu et al (2016)), and theoretical statements are also of interest but reviewers still feel the paper needs improvement in writing and presentation of results. I would recommend an invitation to the workshop track.